

# Graph-based deep fusion for architectural text representation

Shaoyun Hu and Qingxiong Weng

School of Management, University of Science and Technology of China, Hefei, Anhui Province, China

## ABSTRACT

Amidst the swift global urbanization and rapid evolution of the architecture industry, there is a growing demand for the automated processing of architectural textual information. This demand arises from the abundance of specialized vocabulary in architectural texts, posing a challenge for accurate representation using traditional models. To address this, we propose a novel fusion method that integrates Transformer-based models with graph neural networks (GNNs) for architectural text representation. While independently utilizing Bidirectional Encoder Representations from Transformers (BERT) and the robustly optimized BERT approach (RoBERTa) to generate initial document representations, we also employ term frequency-inverse document frequency (TF-IDF) to extract keywords from each document and construct a corresponding keyword set. Subsequently, a graph is created based on the keyword vocabulary and document embeddings, which is then fed into the graph attention network (GAT). The final document embedding is generated by GAT, and the text embedding is crafted by the attention module and neural network structure of the GAT. Experimental results from comparison studies show that the proposed model outperforms all baselines. Additionally, ablation studies demonstrate the effectiveness of each module, further reinforcing the robustness and superiority of our approach.

## INTRODUCTION

With the rapid pace of global urbanization and the swift growth of the architecture industry, there is a rising demand for efficient processing of architecture-related information. Textual information in construction covers everything from planning and design documents for construction projects to building regulations, policy documents, research reports and client requirements (*Wu et al., 2022*). Therefore, it is particularly important to fully utilize various data sources to improve automation and informatization in the architecture industry. In recent years, the construction industry has increasingly utilized natural language processing (NLP) technology. These techniques are used to represent architectural text and serve applications such as recommendation systems, information filtering, document organization, expert systems, and automatic rule checking (*Zheng et al., 2022; Chi et al., 2016*). In the field of architecture, *Fang et al. (2020)* successfully utilized the Bidirectional Encoder Representations from Transformers (BERT) model to extract accident features from historical accident information, achieving detailed

Corresponding author
Qingxiong Weng,
wqx886@ustc.edu.cn

classification and in-depth analysis of accidents, providing powerful decision-making support for enhancing construction safety and work efficiency. *Moon, Lee & Chi (2021)* proposed a construction text matching method based on Doc2Vec, which effectively identifies relevant clauses in different design specifications by calculating cosine similarity, providing a convenient way for contractors to quickly locate required specification clauses.

Word vector models like Word2Vec capture basic contextual information from architectural texts but lack attention to context-awareness. Text Graph Convolutional Networks (TextGCN) have been proposed to address this limitation by constructing a graph-based representation of texts, allowing for the modeling of relationships between words and documents in a non-sequential manner (*Yao, Mao & Luo, 2019*; *Dai et al., 2022*). TextGCN treats all words as nodes in the graph, which can dilute the importance of specialized architectural vocabulary, leading to a loss of crucial relationships between domain-specific terms and documents. Pre-trained language models like BERT, XLNet, and the robustly optimized BERT approach (RoBERTa) (*Devlin et al., 2018*; *Yang et al., 2019*; *Liu et al., 2019*) effectively address these limitations by leveraging extensive prior knowledge, thereby offering a robust context-aware representation of texts that effectively compensates for these shortcomings. Due to the limited availability of architectural text data, pre-trained models may face a deficiency in relevant resources during training. This scarcity can hinder the models' ability to accurately and deeply represent architectural texts. In order to solve the above problems, *Wu et al. (2021)* studied different types of graph neural network (GNN) architecture and discussed the strengths and limitations of each architecture, as well as examined their performance in various domains (*Yao, Mao & Luo, 2019*; *Zong & Sun, 2020*; *Liu et al., 2020*; *Xu et al., 2023*). *Yang et al. (2021)* proposed GraphFormers, an architecture that nests GNN components within transformer blocks, enabling iterative fusion of text encoding and graph aggregation for representation learning on textual graphs. *Soni, Chouhan & Rathore (2023)* is a convolutional neural network (CNN) based architecture designed for both binary and multi-class text classification. It extracts n-gram features at both the intra-sentence and inter-sentence levels, utilizing a unique input matrix format and employing two-dimensional multi-scale convolution for feature extraction. In 2021, *Lin et al. (2021)* proposed a new model for text classification called BERT-GCN. This model combines the advantages of the BERT and graph convolutional networks (GCNs) models, using document embeddings generated by BERT as the input nodes of GCN. Through an iterative graph structure approach, BERT-GCN leverages the large amount of data learned during BERT training and enhances the training data through joint learning within the GCN graph structure. But the BERT model's dependency on context and its limited context window may hinder its ability to capture complex concepts in architectural text, especially when crucial information is scattered across different parts of the text. The resultant document embeddings and keyword lists may lack the necessary representativeness, thereby restricting the model's adaptability to specific contexts and architectural terms. In addition, GCN treats all neighboring nodes equally when processing graph data and is unable to assign varying weights to different nodes. This drawback may weaken the flexibility of GCN when

integrated with BERT, limiting its ability to effectively capture crucial nodes and comprehend intricate relationships between nodes.

In this study, we propose a graph-based deep fusion method to address the complexity of specialized terminology in the field of architecture. This method combines the powerful capabilities of two pre-trained language models, BERT and RoBERTa. This approach leverages the semantic understanding capabilities of BERT and RoBERTa, utilizing their distinct masking strategies and diverse training data to capture features that might be missed by a single model (*Briskilal & Subalalitha, 2022*). These two models can accurately generate document embeddings and keyword vocabularies from architectural text. A graph is then constructed based on document embeddings and the keyword vocabulary, which is subsequently fed into GAT. At the same time, GAT adopts an attention mechanism to identify the interdependencies between keywords and generate the final text embedding. This method effectively combines the context awareness capabilities of BERT and RoBERTa and the attention allocation mechanism of GAT, significantly improving the clarity of architectural text expressions, and improving the performance of the pre-trained model in professional fields. It addresses the limitations of TextGCN by leveraging BERT and RoBERTa to construct document and keyword representations, thereby enhancing the associations between keyword nodes and documents in heterogeneous graphs. In the analysis of architectural text, the graph attention network precisely allocates importance to different nodes, a crucial factor in this process.

In conclusion, the main innovations of this study include:

- We integrate the collaborative characterization methods of BERT and RoBERTa, leveraging their semantic understanding capabilities to model diverse features in architectural texts. Employing the difference masking approach of these two models, we construct document embedding and keyword vocabulary.
- We innovatively use document embeddings and keyword vocabularies to build a graph neural network, which is then fed into GAT. Through the node attention mechanism in GAT, we can identify the interdependencies between keywords and finally generate text embeddings.
- Due to the lack of specialized datasets in the field of architecture, we annotated the gathered architectural texts, resulting in a architecture dataset comprising 13 categories and 20,013 documents.

The remainder of this article is organized as follows: "Related Work" delves into related work in detail, covering BERT, RoBERTa, and GNN. "Problem Definition" will clarify the problem definition and explain the goals and tasks of this research. "Framework Methodology" will delve into discussing the model and method we proposed, including model architecture, algorithm flow, and implementation details. "Experiment" will describe the experimental settings and results, verifying the effectiveness and superiority of our model through comparative experiments. Finally, "Conclusion and Future Work" will summarize the research conclusions and outline future research directions and prospects.

## RELATED WORK

### BERT

BERT, a pre-trained NLP model introduced by Google in 2018, utilizes the Transformer architecture, known for its self-attention mechanisms. To predict the sentiment polarities for the given aspects or targets, *Li et al. (2020a)* use gating mechanisms and context-aware aspect embedding to enhance and control the BERT representation of aspect-based sentiment analysis. *Yang et al. (2020)* leveraged BERT to encode captions and videos, enabling them to answer questions related to the visual content of an image or video. *Moon, Chi & Im (2022)* further employ a BERT-based utterance classification model to identify risk information in each utterance within a architecture specification. *Kim et al. (2022)* employed BERT to extract pertinent information from architecture codes, aiming to develop an information retrieval chatbot in alignment with building codes. *Gao et al. (2019)* harnessed BERT's context-awareness for conducting sentiment analysis on complex sentences within aspect-based sentiment analysis.

### RoBERTa

RoBERTa is a Transformer-based pre-trained model that has a series of enhancements over BERT, including eliminating the next sentence prediction task (*Liu et al., 2019*). *Liu et al. (2022)* used RoBERTa to extract entities and relationships from open-source intelligence for constructing a knowledge graph in a specific domain. They validated their approach using different text similarity algorithms and classification methods. *Ahammad et al. (2024)* introduces the RoBERTa-GCN model, which combines RoBERTa with a GNN, to effectively address the issue of fake news detection in the Bengali language. *Kong et al. (2024)* adopts RoBERTa and Global Vectors for Word Representation (Glove) for global vector representation, solving the challenge of automatically detecting cyberbullying in tweets. *Li et al. (2020b)* introduced a RoBERTa-based word embedding model using the power grid *corpus*, applying it to the task of entity recognition. In addition, *Dai et al. (2021)* employed fine-tuning with RoBERTa to conduct sentiment analysis on English sentences characterized by intricate grammatical structures.

### Graph neural networks

GNNs are connectionist models designed to capture dependencies and relationships among graph nodes by facilitating message passing through the edges connecting these nodes (*Scarselli et al., 2008*). *Yao, Mao & Luo (2019)* proposed using text graph convolutional networks for text classification, which builds a unified text graph based on word co-occurrence and document-word relations. Research has also explored the integration of BERT's semantic understanding capabilities with the graph structure of GCN to tackle diverse tasks. BERTGCN integrates traditional BERT and graph convolutional network (GCN) models. In the BERTGCN model, BERT serves as an embedding layer to represent documents at a foundational level and integrates them into GCN as the node representation. This approach combines the deep semantic feature extraction capabilities of BERT with the ability of GCN to capture relationships in structured data, such as graph structures (*Xue, 2023*; *Ding et al., 2022*; *Yang & Liu, 2023*;

*Xu et al., 2023*). *She, Chen & Chen (2022)* employed a joint model that combines BERT and GCN for text classification tasks. Similarly, *Xiao et al. (2021)* employed a joint model for emotion classification, leveraging BERT's semantic understanding capabilities and GCN's strengths in capturing relationships between nodes. *Kong et al. (2024)* propose a GNN approach for building function classification that integrates multi-source data and mines contextual information between buildings. *Wang, Sacks & Yeung (2022)* introduces a novel approach to semantic enrichment of building information modeling (BIM) models using graph representation and GNNs. *Wang, Sacks & Yeung (2022)* use the room classification task to develop, test and illustrate a novel approach to semantic enrichment of BIM models —representation of models as graphs and application of graph neural networks (GNNs). These studies not only expand the application of graph neural networks in the field of natural language processing but also provide new ideas and methods for solving complex text analysis problems.

# PROBLEM DEFINITION

In this section, we first introduce the semantic fusion. Then we formally define the process of graph-based semantic enhancement.

## Semantic fusion

Our principal objective is to conduct an effective methodology for architectural text representation, with a specific focus on enhancing semantic richness. The texts are denoted as $T_a = \{a_1, a_2, \ldots, a_m\}, a_i \in A$, $m$ refers to the number of architectural texts. Our methodology involves concurrently inputting each text into BERT and RoBERTa to obtain two sets of token embeddings. The text embedding, denoted as $u_{bert}$, is derived by computing the mean pooling of token embeddings obtained from BERT. Similarly, the text embedding, denoted as $v_{roberta}$, is obtained by applying mean pooling to the token embeddings acquired from RoBERTa. Subsequently, the integration of these two disparate semantic representations yield the composite embedding denoted as $e_{composite}$.

## Graph-based semantic enhancement

A large heterogeneous text graph is constructed, which includes keyword nodes and document nodes. The extraction of keyword nodes utilizes the term frequency-inverse document frequency (TF-IDF) method, which calculates the term frequency (TF) of each term in the document and the inverse document frequency (IDF) across the entire *corpus*, and then multiplies these two values to obtain the TF-IDF score. The terms with the highest scores are selected as keywords. The weight of the edges is obtained by calculating the PMI value between pairs of words, which is calculated using the formula $PMI(i, j) = \log \frac{p(i,j)}{p(i)p(j)}$, where $p(i, j)$ is the probability of words $i$ and $j$ co-occurring, and $p(i)$ and $p(j)$ are the probabilities of words $i$ and $j$ occurring, respectively.

Keyword nodes can be represented as a set $W = \{w_1, w_2, \ldots, w_{n_1}\}$, where $w_i$ is a term in the vocabulary, and $n_1$ is the size of the vocabulary. Document nodes can be represented as a set $D = \{d_1, d_2, \ldots, d_{n_2}\}$, where $d_j$ is a document in the dataset, and $n_2$ is the size of the dataset. The number of nodes in the text graph, denoted as $|V|$, $|V| = n_1 + n_2$. This graph

can be represented by an adjacency matrix $A \in R^{|V| \times |V|}$, where $A_{ij}$ represents the weight of the edge between nodes $i$ and $j$.

We input this heterogeneous graph into GAT. GAT can capture and learn the complex relationships between nodes in the graph. A key feature of GAT is its ability to learn the weights of the edges in the graph. This attention mechanism is based on the features of the neighbor nodes $h_j$ and the current node $h_i$. Nodes aggregate the features of neighbor nodes according to these weights $a_{ij}$, generating their new features $h'_i$. In this way, GAT can effectively capture and learn the complex structure of the text graph, and finally output a text embedding that is richer in semantics.

# FRAMEWORK METHODOLOGY

In this section, we propose the graph-based deep fusion model, which is composed of semantic fusion module and graph-based semantic enhancement module. First, the semantic fusion module includes BERT and RoBERTa. Compared with using BERT or RoBERTa alone, our semantic fusion module combines the advantages of both and can extract richer and more comprehensive textual semantic information. The new Architectural text embeddings are inputted into the GAT network, yielding the comprehensive representation of the text embeddings.

As illustrated in Fig. 1, we leverage both BERT and RoBERTa models to obtain text embeddings, which are then concatenated to form a unified representation. By capitalizing on the differences between RoBERTa's dynamic masking mechanism and BERT's next sentence prediction (NSP) task, we effectively capture the latent features of architectural text. Additionally, we construct a specialized vocabulary for the architectural domain, followed by the application of TF-IDF to extract document keywords. Only the keywords that match entries in the pre-established vocabulary, along with their corresponding TF-IDF values, are retained to create a keyword list for each document. Using the document representation and keyword table generated from the two models, a graph is constructed. In this graph, keyword nodes are linked to document nodes based on their TF-IDF values within the document, forming edges between related terms and documents. The resulting graph is then processed by a GAT that employs an attention mechanism to refine these connections, ultimately generating the final, enriched text representation.

## Semantic fusion module

In the semantic fusion module, we propose a novel approach for constructing robust embeddings for architecture texts by leveraging the strengths of both BERT and RoBERTa models. The text matrix of architecture text $T_a$ is composed by architecture text:

$$T_a = \{a_1, a_2, \ldots, a_m\}, a_i \in A \tag{1}$$

where $a_i$ refers to the i-th Architectural text, $m$ refers to the number of Architectural texts.

Utilizing BERT to generate embeddings for a given text $a_i$ involves the initial step of feeding the $a_i$ through the BERT model, which produces contextualized word embeddings. Following this, the mean-pooling operation is applied to these embeddings across the entire sequence, yielding a singular vector representation $u^i_e$ that encapsulates the

**Peer**J Computer Science

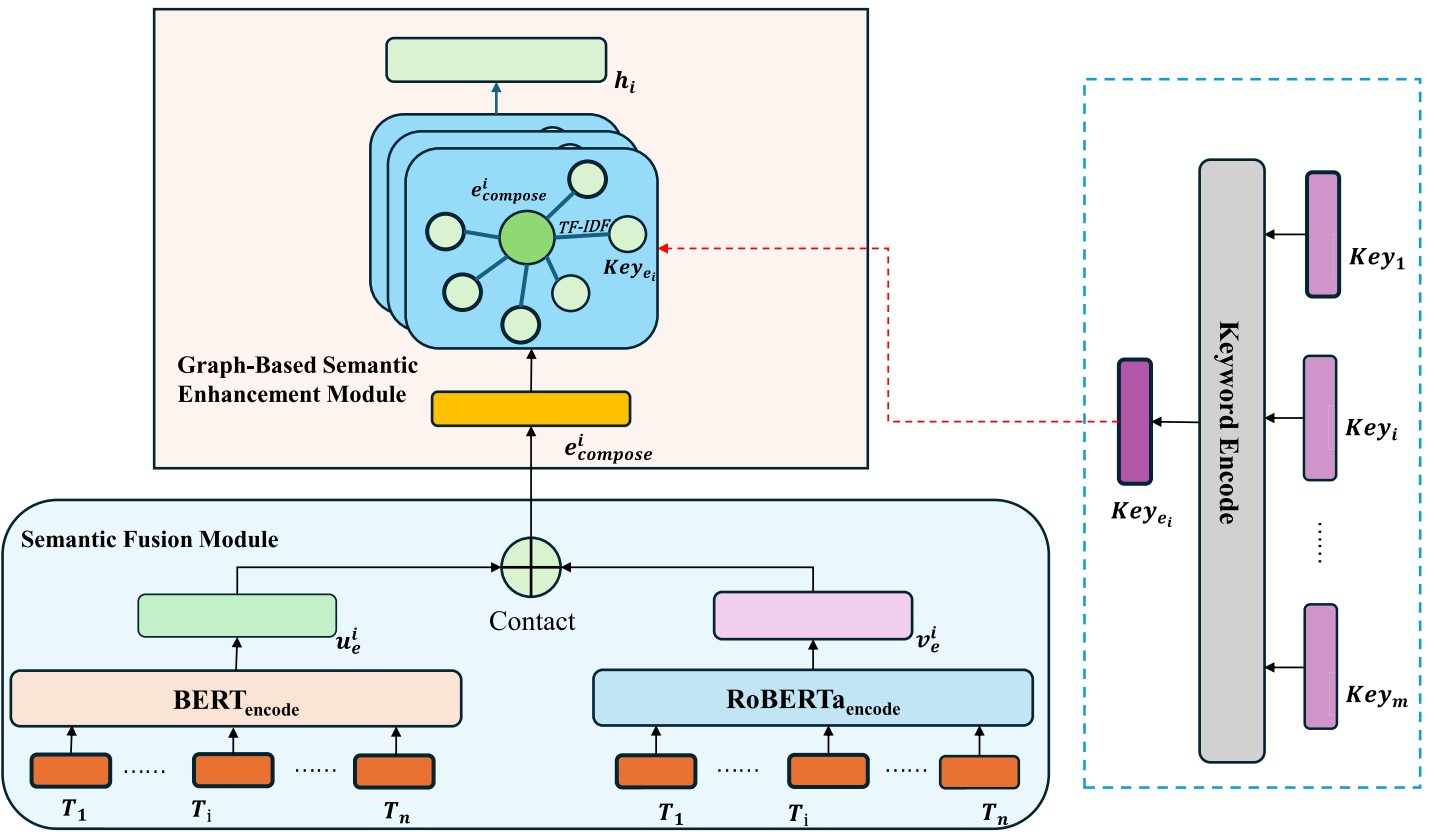

**Figure 1** Graph-based deep fusion model.

aggregated semantic information of the input text. we refer to this function as $f_1$. This resultant embedding $u_e^i$ serves as a condensed representation of the original text $a_i$.

The process of obtaining text embeddings through BERT can be formulated by the following equation:

$$
\begin{aligned}
E_{bert} &= F_1(T_a) \\
&= \{f_1(a_1), f_1(a_2), \ldots, f_1(a_m)\} \\
&= \{u_e^1, u_e^2, \ldots, u_e^m\}, a_i \in T_a
\end{aligned}
\tag{2}
$$

where $u_e^i$ is the embedding of Architectural text $a_i$, $m$ refers to the number of Architectural texts.

Leveraging RoBERTa for text embedding involves initially processing the input text $a_i$ through the RoBERTa model, generating contextualized word embeddings. Unlike BERT, RoBERTa adopts a masked language model objective during pretraining, facilitating a deeper understanding of contextual relationships. The embeddings are then subjected to the subsequent pooling operation, such as mean-pooling across the entire sequence. We refer to this function as $f_2$. Similarly, this operation also results in a singular vector representation $v_e^i$, capturing the aggregated semantic information of the input text. This resultant embedding $v_e^i$ serves as a contextually rich representation of the Architectural

text. The process of obtaining text embeddings through BERT can be formulated by the following equation:

$$
\begin{aligned}
E_{roberta} &= F_2(T_a) \\
&= \{f_2(a_1), f_2(a_2), \ldots, f_2(a_m)\} \\
&= \{v_e^1, v_e^2, \ldots, v_e^m\}, a_i \in T_a
\end{aligned}
\tag{3}
$$

where $v_e^i$ is the embedding of architectural text $a_i$, $m$ refers to the number of architectural texts.

Subsequently, the embeddings from both models are concatenated to form a composite textual embedding, providing a more comprehensive representation of semantic information. This can be expressed mathematically as follows:

$$
e_{composite}^i = \text{Concat}(u_e^i, v_e^i)
\tag{4}
$$

$$
E_{composite}^i = \{e_{composite}^1, e_{composite}^2, \ldots, e_{composite}^m\}
\tag{5}
$$

where $e_{composite}^i$ is the fusion embedding of Architectural text $a_i$, $m$ refers to the number of architectural texts.

This hybrid embedding approach leverages the differences between BERT and RoBERTa in text representation processing. By combining BERT's next sentence prediction (NSP) task with RoBERTa's dynamic masking mechanism, and concatenating the resulting vectors, the model enhances its ability to extract semantic features from architectural text.

## Graph-based semantic enhancement module

### Text graph

The goal of this module is to construct a large, heterogeneous text graph that integrates both word nodes and document nodes. This approach aims to explicitly model global word co-occurrences, providing a more comprehensive representation of text relationships. Additionally, it enhances the flexibility of adjusting graph convolution operations. An example illustrating the method used in this article for graph construction is presented in Fig. 2.

To ensure consistency in the keywords extracted by both models when merging the keyword tables generated by BERT and RoBERTa, we opted not to use these models for keyword extraction. Instead, we employed the TF-IDF method. First, preprocessing operations such as word segmentation are performed on the text to create a basic vocabulary list. Subsequently, we compute the TF for each term in the document, as well as the IDF across the entire *corpus*. By multiplying each term's TF and IDF values, we can calculate its TF-IDF score. This score reflects the term's importance in a specific document and its uniqueness in the entire *corpus*.

In order to filter out the most representative keywords, we designed a ranking mechanism to rank terms in the vocabulary based on their TF-IDF scores. Finally, the top five terms with the highest TF-IDF scores are selected as the keywords of the document.

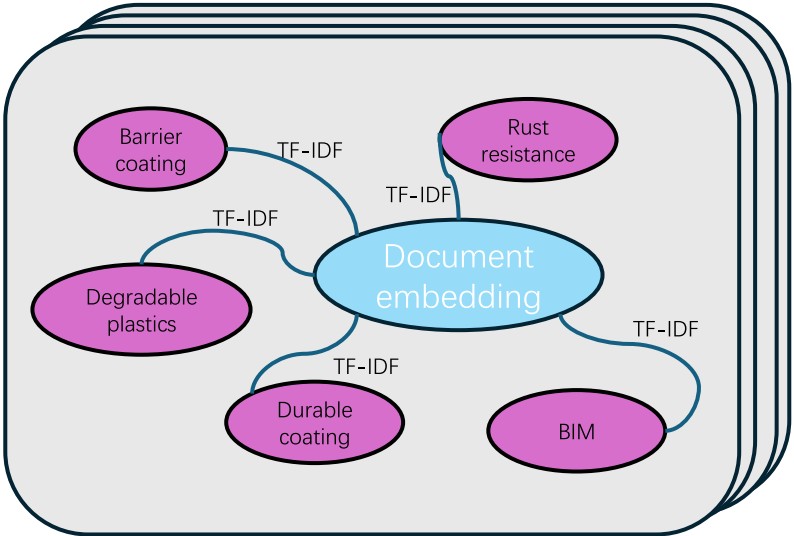

**Figure 2  Text graph.**                               

This approach prioritizes terms that appear frequently in documents but are relatively rare in the entire *corpus*, thus providing a richer and more comprehensive representation of semantic information.

The number of nodes in the text graph, denoted as $|V|$, is equal to the sum of the number of documents in the Architectural text *corpus* and the size of the vocabulary of unique words in the Architectural text. Our methodology involves concurrently feeding each text into both BERT and RoBERTa models, yielding two sets of token embeddings. These text embeddings, denoted as $e_{bert}$ and $e_{roberta}$, are derived by computing the mean pooling of the token embeddings obtained from each model. Subsequently, we concatenate $e_{bert}$ and $e_{roberta}$ to form a combined embedding denoted as $e_{composite}$. Every word and document is represented as $e_{word}$ and $e_{composite}$, which serve as the input for GAT.

We construct edges between nodes based on the occurrence of words in documents (document-word edges) and the co-occurrence of words across the entire *corpus* (word-word edges). The weights of edges between document nodes and word nodes are determined by TF-IDF of words in the documents, where term frequency represents the number of occurrences of a word in a document, and inverse document frequency is the logarithmically scaled inverse of the number of documents containing that word. We found that using TF-IDF weights performs better than using term frequency alone.

In order to fully exploit the global word co-occurrence information, we adopt a fixed-size sliding window to traverse all documents in the *corpus* and collect co-occurrence statistics. When calculating the weight between two word nodes, we use pointwise mutual information (PMI), a popular measure of word association. In our preliminary experiments, we observed that employing PMI yielded superior results compared to using word co-occurrence counts.

Formally, the weight of the edge between nodes $i$ and $j$ is defined as:

$$A_{ij} = \begin{cases} PMI(i,j) & \text{if } i, j \text{ are words and } i \neq j \\ TF - IDF_{ij} & \text{if } i \text{ is a document and } j \text{ is a word} \\ 1 & \text{if } i = j \\ 0 & \text{otherwise} \end{cases} \quad (6)$$

TF-IDF The PMI value of a word pair i, j is computed as:

$$\begin{aligned} PMI(i,j) &= \log \frac{p(i,j)}{p(i)p(j)} \\ p(i,j) &= \frac{\#W(i,j)}{\#W} \\ p(i) &= \frac{\#W(i)}{\#W} \end{aligned} \quad (7)$$

where $\#W(i)$ represents the number of sliding windows in the *corpus* that contain word i, and W(i,j) represents the number of sliding windows in which both word i and j co-occur. Positive PMI values indicate higher semantic relevance between words in the *corpus*, while negative PMI values indicate low or no semantic relevance. Therefore, we only add edges between word pairs that have positive PMI values.

In the graph, nodes D4, D23, and D101 correspond to the labels assigned to architectural texts. other words represent the keywords extracted from the text. The black lines indicate the connections between the articles and the keywords, whereas the orange line represents the connections between the keywords and the articles. The complete configuration constitutes a graph structure, which serves as the input for the GAT model.

### GAT

In the architectural text (*Pal, Selvakumar & Sankarasubbu, 2020*), for any two nodes $i, j$, the $e_{\text{word}}$ and $e_{\text{composite}}$ are used as the initial node feature representations, respectively, and the two are spliced together, and then the inner product is computed with a weight vector $\alpha$ to derive the attention coefficient $\alpha_{ij}$ used to measure the importance of the relationship between node $i$ and node $j$. The formula is calculated as follow:

$$\alpha_{ij} = \frac{\exp\left(\text{LeakyReLU}\left(\alpha^T[wh_i||wh_j]\right)\right)}{\sum_{k \in N_i} \exp(\text{LeakyReLU}(\alpha^T[wh_i||wh_k]))} \quad (8)$$

In the above equation, $N$ is the total number of nodes, $||$ is the splicing operation, denoted as transpose, $w$ is the weight matrix, and $k$ is one of the $N$ nodes. *LeakyReLU* is an activation function that allows a small, non-zero gradient for negative inputs, preventing the "dying ReLU" problem by introducing a slight slope for negative values. The feature vector of node $i$ after the attention mechanism can be expressed as:

$$h_i' = W_a\left(\sum_{j \in N_i} \alpha_{ij} W h_j\right) \quad (9)$$

$W_a$ is the corresponding weight matrix, $h_i$ is the feature vector of node i and $h_j$ is the feature vector of node j. In order to stabilize the learning process of self-attention, we incorporate the use of multi-head attention. In our model, K attention mechanisms (K = 8

in the study) are employed to perform the transformation described in the equation above. The resulting features from these K attention mechanisms are then concatenated. For the final layer of the network, we apply multi-head attention using an averaging approach. The outputs of the K attention mechanisms are processed through an aggregation function, and the output of the final attention layer as $h_i^*$:

$$h_i^* = \frac{1}{K} \sum_{k=1}^{K} W_a \left( \sum_{j \in N_i} \alpha_{ij}^k W^k h_j \right). \tag{10}$$

In Fig. 3, node $h_i$ is connected to its neighboring nodes, represented by node $h_1, h_2, h_3, h_4, h_5$. The attention coefficients between node $i$ and its neighboring nodes are denoted by $\alpha_{i1}, \alpha_{i2}, \alpha_{i3}, \alpha_{i4}, \alpha_{i5}$. The graph in the illustration only shows a single-head attention mechanism.

Finally, We can get the embeddings of the architectural text, ultimately contributing to improved downstream tasks and applications.

## Time and space complexity analysis

For the combined BERT and RoBERTa model, the time complexity is primarily determined by the self-attention mechanism and the feedforward neural network layers. The time complexity can be expressed as:

$$T_{\text{BERT + RoBERTa}} = O\big(L_{\text{BERT}} \cdot (n^2 \cdot d + n \cdot d^2) + L_{\text{RoBERTa}} \cdot (n^2 \cdot d + n \cdot d^2)\big)$$

where $L_{\text{BERT}}$ and $L_{\text{RoBERTa}}$ denote the number of layers in BERT and RoBERTa, $n$ is the input sequence length, and $d$ is the embedding dimension. The self-attention mechanism contributes a complexity of $O(n^2 \cdot d)$, while the feedforward neural network adds a complexity of $O(n \cdot d^2)$.

Similarly, the space complexity for the combined BERT and RoBERTa model, which includes the storage of model parameters and intermediate activations, is given by:

$$S_{\text{BERT + RoBERTa}} = O\big(L_{\text{BERT}} \cdot (d^2 + n \cdot d) + L_{\text{RoBERTa}} \cdot (d^2 + n \cdot d)\big)$$

where $L_{\text{BERT}}$ and $L_{\text{RoBERTa}}$ represent the number of layers, $d$ is the embedding dimension, and $n$ is the sequence length.

For the GAT (Graph Attention Network), the time complexity is largely influenced by the number of nodes and edges in the input graph and the attention mechanism used during graph convolution. The time complexity for a single layer of GAT is:

$$T_{\text{GAT}} = O\big(L_{\text{GAT}} \cdot (|V| \cdot d^2 + |E| \cdot d)\big)$$

where $L_{\text{GAT}}$ represents the number of GAT layers, $|V|$ is the number of nodes, $|E|$ is the number of edges in the graph, and $d$ is the feature dimension of each node. The $O(|V| \cdot d^2)$ term arises from the dense transformation of node features, while the $O(|E| \cdot d)$ term stems from the attention mechanism, which computes attention coefficients for each edge.

The space complexity for GAT involves storing node features, edge features (attention coefficients), and intermediate activations. It is expressed as:
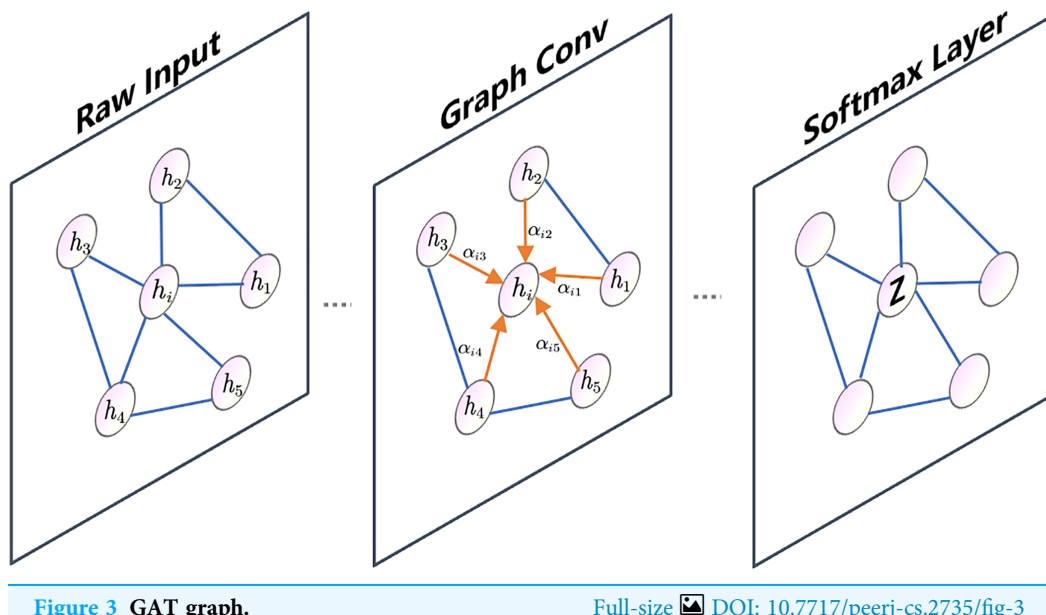

**Figure 3 GAT graph.**

$$S_{GAT} = O(L_{GAT} \cdot (|V| \cdot d + |E|))$$

where $|V|$ and $|E|$ represent the number of nodes and edges, respectively, and $d$ is the dimension of the node embeddings. The term $|V| \cdot d$ accounts for storing node embeddings, and $|E|$ corresponds to the space needed to store edge-wise attention coefficients.

## EXPERIMENT

We conduct relevant experiments to demonstrate the effectiveness of our proposed method. Specifically, we first describe the architecture domain dataset and the generic news dataset constructed in the experiments. In addition, we introduce the concepts related to the baseline model. During the experimental modeling process, we describe the training setup in detail, including dataset partitioning. Finally we conduct experiments in the following two aspects: (1) the performance of our model was compared with other methods on the architecture dataset and the news dataset. An ablation study was conducted to analyze the contribution of each component of the model; (2) verification through t-SNE dimensionality reduction experiments improved the clustering performance of the model.

### Datasets

Due to the lack of professional dataset at the architecture domain, we collected text data from professional websites at the architecture field through crawlers and other techniques. Meanwhile, we invited experts in the domain of a total of 20,013 architecture to guide and label the dataset, which contains 13 categories and data. These categories cover a variety of topics, including architecture materials, science, and architecture appreciation. To measure the model's ability to generalize over generic dataset, we used generic news dataset to evaluate its performance. The generic news dataset contains 18,569 news articles from

**Table 1 Statistics of the architecture dataset and news dataset.**

| Data | Statistics | Architecture | News |
|---|---|---|---|
| Text | # of documents | 20,013 | 18,569 |
| Label | # of labels | 13 | 14 |

2005 to 2011. It covers 14 different categories including various topics such as sports, finance, and stocks. The architecture dataset and news dataset are shown in Table 1.

## Training settings

Before training the model, the two dataset are randomly divided into a training set and a test set in the ratio of 8:2. In order to ensure that the model can converge stably and achieve optimal performance, we adopt the Adam optimizer and set the batch size to 5 and the learning rate to 0.00005. We also implement algorithms including dropout and regularization to avoid over-fitting, which generates a slight improvement in model performance and thus will not be discussed in detail. On both dataset, we train our models for at most 60 epochs to obtain the best performances. In the experiments, the graph is constructed using documents and keywords as nodes. Keyword nodes are linked to document nodes based on their occurrence within the documents, and keyword nodes are interconnected based on the semantic similarity of their embeddings. This structure captures both document-level and term-level relationships within the text.

## Baseline

To evaluate the effectiveness of graph-based deep fusion method, this article designed comparative experiments including Word2Vec, XLNet, SentenceTransformer, Bert, BERT+GCN and TextConvoNet. In evaluating the representational capabilities of each model, the performance of each embedding method was chosen to be judged with a text categorisation task and accuracy, precision, recall and F1-score were used as evaluation metrics. The baseline models are described in detail below.

- Word2Vec (*Church, 2017*): Word2Vec utilizes a shallow neural network to train word embeddings.
- XLNet (*Yang et al., 2019*): XLNet combines the self-attention mechanism of the Transformer with the benefits of an autoregressive language model. It learns bidirectional context relationships of words through a permutation language modeling approach.
- SentenceTransformer (*Reimers, 2019*): SentenceTransformer is a BERT-based framework specifically designed to generate sentence-level embeddings. It modifies the target function and the structure of BERT to produce embeddings that are more suitable for comparing sentences.
- BERT (*Devlin et al., 2018*): BERT is a deep learning model that uses the Transformer for pre-training language representations. It learns word embeddings by considering the context on both sides of the sentence simultaneously.

**Table 2 Performance of representation models on the architecture dataset and news dataset.** Bold indicates the highest data in this column.

| Methods | Architecture | | | | News | | | |
|---|---|---|---|---|---|---|---|---|
| | Accuracy | Precision | Recall | F1-score | Accuracy | Precision | Recall | F1-score |
| Word2vec | 0.6041 | 0.5651 | 0.5280 | 0.5091 | 0.9002 | 0.8709 | 0.7482 | 0.7700 |
| BERT | 0.5571 | 0.5339 | 0.5035 | 0.5091 | 0.9072 | 0.8253 | 0.8283 | 0.8241 |
| XLNet | 0.5503 | 0.5082 | 0.5084 | 0.4962 | 0.9059 | 0.8219 | 0.8498 | 0.8343 |
| SentenceTransformer | 0.5663 | 0.5143 | 0.4881 | 0.4913 | 0.8799 | 0.8146 | 0.7730 | 0.7905 |
| BERT+GCN | 0.5901 | 0.5746 | 0.5062 | 0.5382 | 0.8547 | 0.8991 | 0.8234 | 0.8663 |
| TextConvoNet | 0.6011 | 0.5711 | 0.5544 | 0.5684 | 0.9344 | 0.9017 | 0.8847 | 0.8981 |
| BERT+RoBERTa+GAT | **0.6322** | **0.6091** | **0.5778** | **0.5845** | **0.9539** | **0.9113** | **0.9049** | **0.9081** |

- BERT+GCN (*Lin et al., 2021*): The BERT+GCN method combines GCN and BERT to improve performance to further improve performance through graph structures. It applies BERT to process textual data and produce node embeddings, which are then fed into the GCN. The document embedding is iteratively updated based on the graph structure using GCN, and its output is considered the final embedding of the document node.
- TextConvoNet (*Soni, Chouhan & Rathore, 2023*): TextConvoNet is a CNN-based architecture for binary and multi-class text classification. It captures n-gram features both within and across sentences, using a novel input matrix representation and applying two-dimensional multi-scale convolution for feature extraction.

## Results and analysis

To validate the effectiveness of our proposed method, we first evaluate the model's representational capabilities on the architecture dataset to demonstrate its performance on architecture domain. Subsequently, we tested the model on the news dataset to verify its generality and generalization ability. In the experiments, we selected all the baselines mentioned in "Baseline" for comparison along with the ablation study of the models. Also we used evaluation metrics to evaluate the model performance. Table 2 lists the overall results on both datasets with the evaluation metrics mentioned, and Table 3 lists the results of the ablation study on both datasets.

Table 2 shows the accuracy of various models tested on the architecture domain dataset and the news dataset. In the architecture domain dataset, The graph-based deep fusion method performs the best with 63.22%, 60.91%, 57.78%, and 58.45% in terms of accuracy, precision, recall, and F1-scores of its model, which improves compared to BERT in these four metrics by improving the improvement of 7.51%, 7.52%, 6.87%, and 7.54%, respectively. To further validate the generalisation of the model, it can be seen that on the news dataset, the graph-based deep fusion method achieves a significant performance, with an accuracy of 95.39%, which is significantly higher than that of other models for classification. This highlights the strong applicability of this method. This approach takes advantage of the complex text encoding capability of BERT and RoBERTa, and the

**Table 3 Results of ablation study on the architecture dataset and news dataset.** Bold indicates the highest data in this column.

| Methods | Architecture | | | | News | | | |
|---|---|---|---|---|---|---|---|---|
| | Accuracy | Precision | Recall | F1-score | Accuracy | Precision | Recall | F1-score |
| BERT+GAT | 0.6239 | 0.5797 | 0.5334 | 0.5511 | 0.9537 | 0.8822 | 0.8249 | 0.8499 |
| RoBERTa+GAT | 0.6286 | 0.5833 | 0.5498 | 0.5755 | 0.9535 | 0.8831 | 0.8239 | 0.8511 |
| BERT+BERT+GAT | 0.6221 | 0.5821 | 0.5507 | 0.5788 | 0.9531 | 0.8819 | 0.8239 | 0.8554 |
| RoBERTa+RoBERTA+GAT | 0.6301 | 0.5911 | 0.5571 | 0.5642 | 0.9533 | 0.9019 | 0.8877 | 0.8715 |
| RoBERTa+RoBERTA+GCN | 0.6237 | 0.5977 | 0.5574 | 0.5841 | 0.9477 | 0.9089 | 0.8971 | 0.8924 |
| BERT+RoBERTa | 0.6017 | 0.5674 | 0.5179 | 0.5374 | 0.9175 | 0.8698 | 0.8173 | 0.8074 |
| BERT+RoBERTa+GAT | **0.6322** | **0.6091** | **0.5778** | **0.5845** | **0.9539** | **0.9113** | **0.9049** | **0.9081** |

complex relationship modelling capability of GAT, which enhances the model's ability to capture textual contextual relationships, improves the text modelling capability of BERT and RoBERTa, and ultimately improves the accuracy of text classification. Of course, compared to the news dataset, the metrics of all models decreased on the building dataset, reflecting the problems associated with this highly specialised dataset. Nonetheless, this proposed approach still performs well on all four evaluation metrics. The poor performance of these baselines in the architectural domain highlights the limitations of word embedding models and pre-trained models in capturing the latent features specific to architectural text. Our approach effectively exploits the differences between BERT and RoBERTa by combining them to capture textual features that are ignored by individual models. The attention mechanism introduced by the GAT graph neural network improves the quality of the keyword representations, which in turn improves the robustness and performance of the models.

In addition, we conducted ablation study on the models. The results in Table 3 indicate that the absence of the BERT or RoBERTa model results in a decline in metrics. For instance, in the precision metric, the removal of BERT and RoBERTa models led to decreases of 2.58% and 2.94%, respectively. This is because the combination of BERT and RoBERTa, leveraging their complementary features, allows for a more comprehensive understanding of architectural text. In addition, we removed the GAT module and replaced it with GCN, observing a significant decrease in the mode's performance across various parameters. The drop in performance indicators highlights the critical role of GNN in generating the final document representation, with GAT demonstrating the most substantial impact on improving the model. The attention mechanism in GAT enables the model to learn accurate representations of keywords from context, leading to a better understanding of relationships between keywords. Moreover, comparative experiments using the graph-based deep fusion method highlight the contribution of the joint representation to the model's ability to represent architectural text, enhancing its robustness and performance. RoBERTa compensates for any shortcomings that BERT may exhibit on specific architectural data. The ablation study emphasize the importance of each

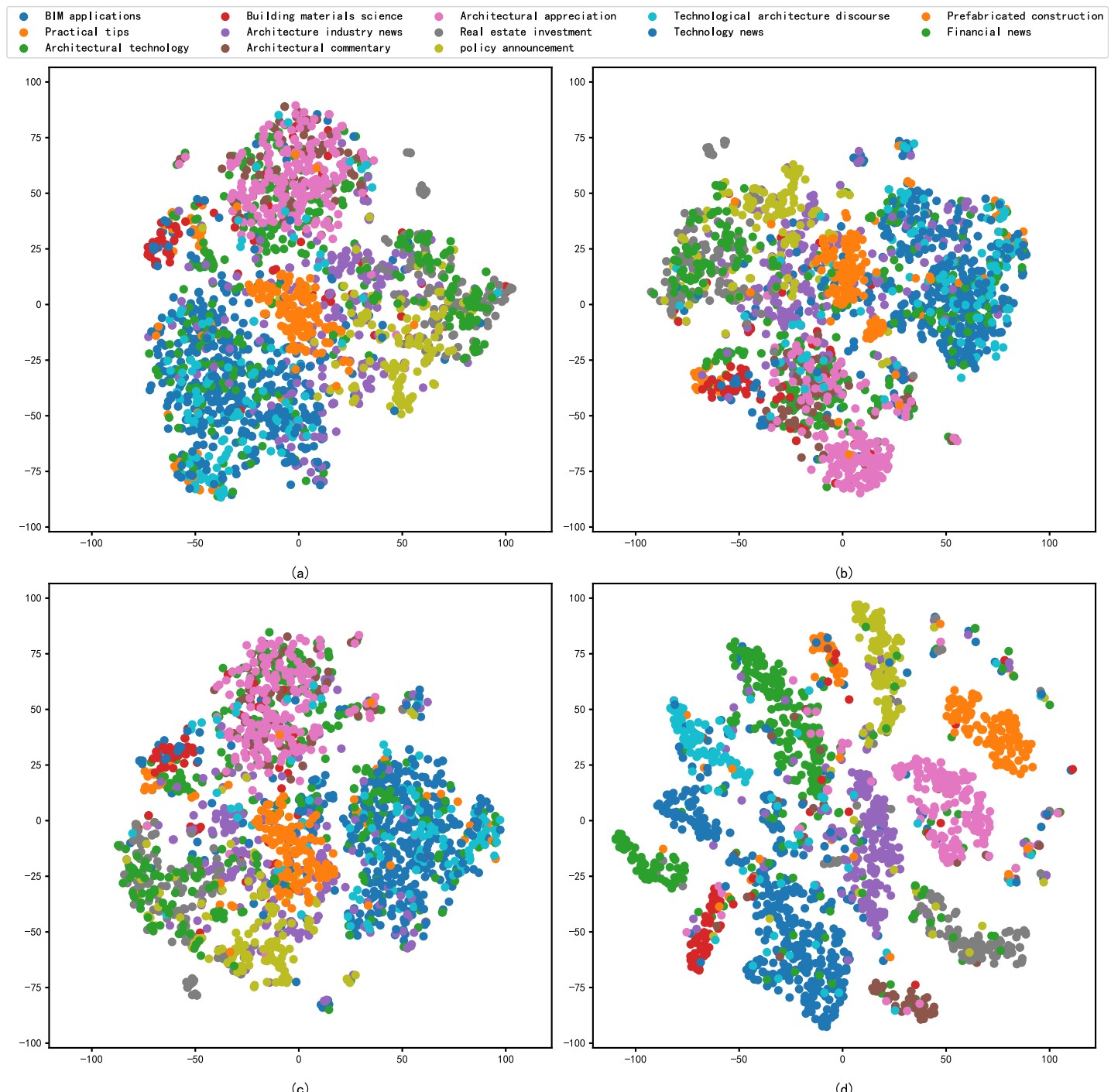

**Figure 4 t-SNE visualization results of different models on the architecture dataset: (A) Word2vec, (B) BERT, (C) xlnet, (D) graph-based deep fusion method.**

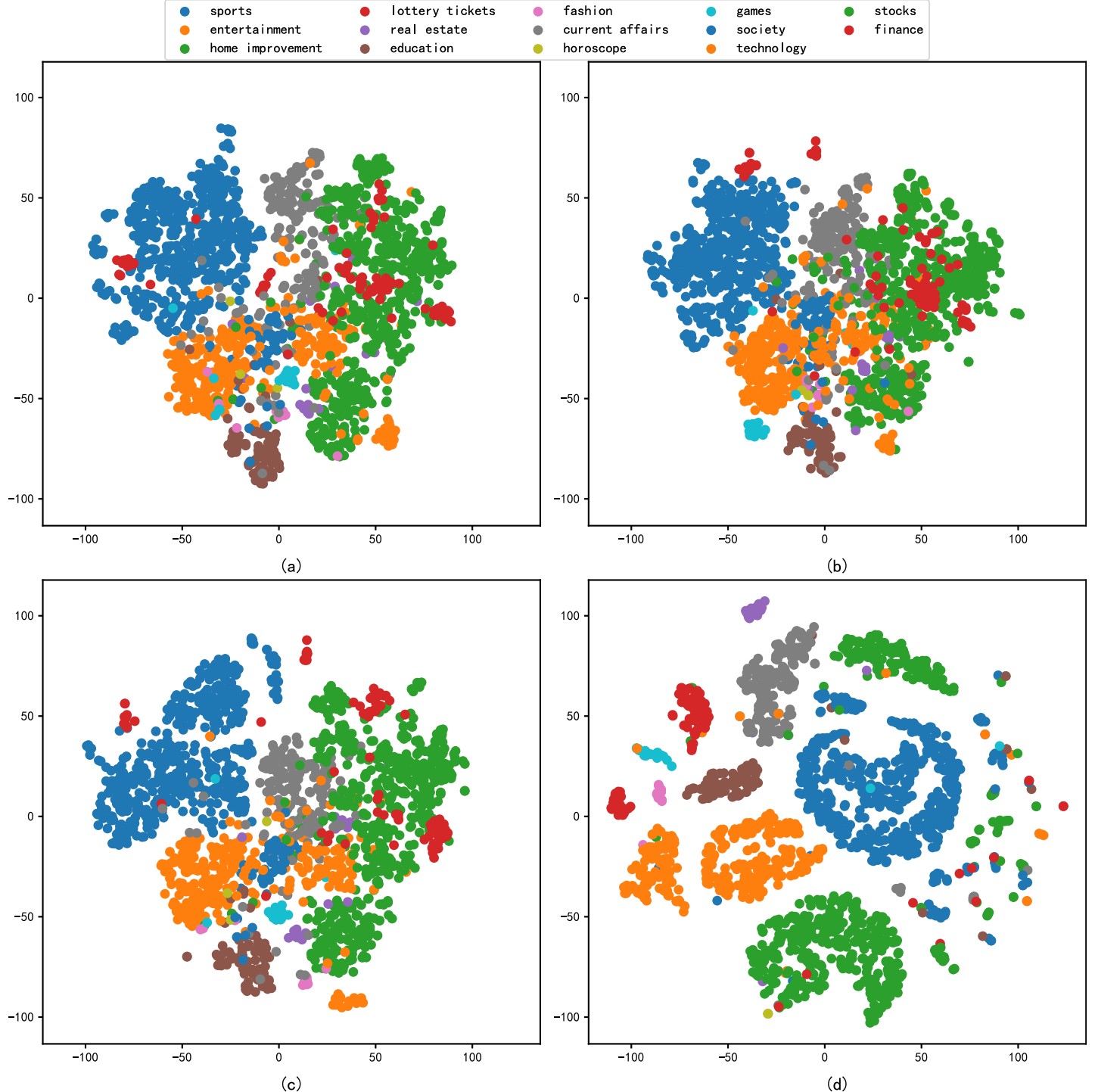

**Figure 5** t-SNE visualization results of different models on the news dataset: (A) Word2vec, (B) BERT, (C) Glove, (D) graph-based deep fusion method.

component in the proposed method structure, as the removal of either BERT or RoBERTa results in a decline in evaluation metrics across all aspects.

## Analysis of reduced dimension

In addition, this study includes the reduced dimension by t-SNE for the different embedding methods, followed by a comparative analysis through the visualisation of scatter plots. The primary criterion for evaluation is the degree of clustering within the same categories in the scatter plots. The results of the performance of each model in the architecture dataset after reduced dimension are shown in Fig. 4, and the results of the performance of each model in the news dataset after reduced dimension are shown in Fig. 5.

As can be seen from Fig. 4, the sentence representations in Fig. 4D are clearly optimal compared to Figs. 4A–4C, where the samples of different architecture document are apparently separated from each other. These models, having not been explicitly trained on architecture domain data or having limited training samples in this domain, exhibit poor text representation capabilities. In contrast, we propose a text representation method that combines RoBERTa and BERT. This approach enhances the model's comprehension of architectural texts and effectively extracts keywords to represent articles. These keywords are subsequently processed with the assistance of GAT. Leveraging the attention mechanism of the nodes in the graph neural network, the model captures relationships between keywords, resulting in an improved representation of the text.

As shown in Fig. 5, we compare the four different model feature representation capabilities in a generic news dataset by t-SNE visualisation. In Figs. 5A–5C, samples from different embedding models are poorly separated, while in Fig. 5D, samples from different news categories are densely clustered together. It shows that the results of feature representation of the model constructed by the method proposed in this article are significantly better than those in Figs. 5A–5C. It also shows that the feature representation capability of the model can effectively learn architectural texts information features with good representability.

## CONCLUSION AND FUTURE WORK

In conclusion, we propose a graph-based deep fusion method for architectural text representation and achieve significant progress. This method integrates the collaborative representation capabilities of BERT and RoBERTa, enhancing semantic understanding in architectural text through their complementary properties. Addressing limited dataset, an extensive search for relevant text resulted in 13 labels and 20,013 documents. Looking forward, future work could explore the incorporation of external knowledge to enhance the semantic richness and focus on constructing a more comprehensive dataset for robust analyses in architectural text representation. Additionally, we aim to integrate various architectural resources, such as architectural design drawings, presentation videos, and engineering conference audio, to further enhance our model. By enriching the diversity of these resources, our model could better support downstream tasks such as urban planning, building information modeling (BIM) enhancement, and more. Furthermore, applying

this method to datasets from other domains could allow us to explore its cross-domain applicability and generalization ability, verifying the model's robustness and versatility beyond architecture.

### Funding
The authors received no funding for this work.

### Competing Interests
The authors declare that they have no competing interests.

### Author Contributions
- Shaoyun Hu conceived and designed the experiments, performed the experiments, analyzed the data, performed the computation work, prepared figures and/or tables, authored or reviewed drafts of the article, and approved the final draft.
- Qingxiong Weng conceived and designed the experiments, authored or reviewed drafts of the article, and approved the final draft.

### Data Availability
The main code of the model is available in the Supplemental File.

The data of the experimental results are available in the Supplemental File.

The dataset and detailed code are available at GitHub and Zenodo:

- https://github.com/allmight1110/BERT-RoBERTa-GAT.

- allmight1110. (2025). allmight1110/BERT-RoBERTa-GAT: V1.1 (Version v1).
Zenodo. https://doi.org/10.5281/zenodo.14855162.

### Supplemental Information
Supplemental information for this article can be found online at http://dx.doi.org/10.7717/peerj-cs.2735#supplemental-information.

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
