# Peer review of "Graph-based deep fusion for architectural text representation"

_PeerJ Computer Science, doi:10.7717/peerj-cs.2735_

## Round 0.1 · original submission · Major Revisions

Based on the reports from our reviewers and my own assessment as Editor, I am pleased to inform you that your manuscript is potentially acceptable for publication in PeerJ Computer Science, provided that you address all the essential revisions suggested by our reviewers.

Reviewer 1 ·

Basic reporting

In this paper, authors proposed a novel text representation learning through the combination of different transformer-based textual embedding models with text graph construction with graph attention network to learn the rich sematic representations of input texts. Specifically, within the proposed technique in this paper, first the input textual documents are fed into the BERT and RoBERTa model to achieve the local rich-semantic representations of occurring words within each document. Then, the document embeddings are utilized to construct the text graph, similar to the approach of TextGCN model, to facilitate the next-step text representation learning through the GAT-based model. This approach is promising for enhancing the semantic information which can extract from the input textual documents as well as constructed text graphs. Extensive empirical studies within two real-word textual datasets within the architecture and general news domains have proved the effectiveness as well as outperformance of authors proposed techniques in this paper. In my opinion, the proposed ideas within this paper are quite interesting in which making the connection between the transformer-based and graph neural network-based for dealing with rich-semantic and global contextual textual representation learning. Beside good points of this paper, I also have some revision recommendations as well as questions about the methodology of this paper for authors, including:
1) First of all, the contents within abstract and introduction sections of this paper should be revised in order to more highlight on the main approach of this paper in which the combination between transformer and GNN-based text graph embedding approach. More specify on how their current study/approach is different from previous techniques, such as TextGCN, and how these differences can help to achieve better performance of text representation learning/classification problem?
2) More pros/cons of previous text embedding techniques should be discussed within the literature review of this paper, and how the drawbacks of previous techniques can be resolved/overcome through authors’ proposals in this paper?
3) For the process of text representation learning, through BERT/RoBERTa models – as described within methodological section, why these transformer-based techniques are selected, what are the research assumptions of authors on the use of these transformer-based techniques?
4) Related to the method of building the text graph, why the TF-IDF is applied in this case, as we also have the word embedding vectors have been learnt through the BERT/RoBERTa models – why authors don’t directly use these feature representations? And what’s main differences of authors’ text graph construction technique in comparing with previous TextGCN model? Please explain more about that.
5) There are also many mathematical notations/terms are utilized without first-time explanations, should I suggest authors to carefully check, revise and adding missing descriptions for these mathematical notations/terms.
6) Finally, as there are multiple deep learning-based architectures have been utilized within authors’ proposed technique in this paper, thus I thought additional time/space complexity analysis might be necessary to have within the last part of the methodological section.

Experimental design

No comment.

Validity of the findings

Please refer to my basic reporting section.

Additional comments

No comment.

Cite this review as

Reviewer 2 ·

Basic reporting

In this paper, the authors propose a graph-based deep fusion method for architectural text representation by leveraging Graph Attention Networks (GAT) alongside BERT and RoBERTa. The approach aims to address the challenges posed by the specialized vocabulary in architectural texts by combining keyword and document embeddings. This innovative method is particularly intriguing, and I would appreciate further clarification on several aspects of your methodology and results.

Experimental design

1) Can you elaborate on how BERT and RoBERTa representations are combined in your fusion method?

2) What were the key considerations in choosing these specific models for architectural text representation?

3) How does the fusion of keyword and document embeddings enhance the model’s performance compared to using only one type of embedding?

4) What challenges did you encounter when applying Graph Attention Networks (GAT) to architectural text processing, and how did you address them?

5) Could you explain in more detail how the graph based on keyword vocabulary and document embeddings is constructed? What criteria are used to connect nodes in the graph?


6) Could you provide more details about the baseline models used for comparison in your experiments? How do they differ from your proposed method in terms of architectural text processing?
In which specific areas or tasks did your proposed model show the most significant improvements over the baselines?

7) In the ablation studies, which specific modules contributed the most to the overall performance of the model? Were there any surprising findings regarding module effectiveness?

8) Can you describe how the removal of certain components affected the model’s performance, particularly in relation to the attention mechanism or the graph construction?

Validity of the findings

9) What are the next steps in refining or expanding your approach? Are there other areas within architecture or similar domains where you see potential applications for this model?


10) Have you considered expanding your approach to multimodal architectures, where text is combined with visual or spatial data?

11) Additionally, I suggest incorporating recent references.

Cite this review as

Reviewer 3 ·

Basic reporting

1. Specialized terms and acronyms, such as GCN, should be introduced with their full names when first mentioned in the manuscript.
2. The related work section should also include a more comprehensive survey of research on architectural text processing, especially other works employing GNNs and other deep learning algorithms for this task.

Experimental design

The proposed model incorporates BERT and RoBERTa as part of its backbone, and this should be described in greater detail. It is recommended to include illustrations to enhance clarity and understanding.

Validity of the findings

The baseline models compared in Table 2 should be clearly referenced with their sources, and the corresponding citations should be provided. It is also suggested to include some of the latest works in the baseline comparison.

Additional comments

None

Cite this review as

---

## Round 0.2 · accepted · Accept

Based on the reviewers' reports and my own assessment as Editor, I am pleased to inform you that your manuscript is acceptable for publication in PeerJ Computer Science, provided that the authors address the minor concerns raised by Reviewer 3:

Include specific experimental results in the abstract, such as accuracy improvements or numerical values, to enhance its persuasiveness.
Add relevant keywords to the manuscript.

Ensure that all mathematical notations and terms are clearly explained upon first use.

Resolve outstanding abbreviation inconsistencies by carefully reviewing and revising the text.

Reviewer 1 ·

Basic reporting

After thoroughly reviewing the revisions made in the latest version of the manuscript and the authors' responses to the reviewers' comments, I confirm that all issues identified in the previous version have been adequately addressed. I believe the manuscript is now suitable for publication in its current form. Thank you.

Experimental design

No comment.

Validity of the findings

Please refer to my basic reporting section.

Additional comments

No comment.

Cite this review as

Reviewer 3 ·

Basic reporting

The manuscript has been revised according to the review comments. However, there are still some issues that need to be addressed:

1. I suggest using specific experimental results in the abstract, such as the degree of accuracy improvement or numerical values, to increase its persuasiveness.

2. The manuscript lacks keywords.

3. There are also many mathematical notations/terms are used without first time explanations, I suggest authors to carefully check and revise.

4. There are also many abbreviation problems have not been resolved, I suggest authors to carefully check and revise.

The authors should carefully address these remaining issues before the manuscript can be considered for publication. I recommend another round of revision to further improve the quality and clarity of the manuscript.

Experimental design

no comment

Validity of the findings

no comment

Additional comments

no comment

Cite this review as